**www.cambridge.org/ext**

law and governance; preventing extinction; declaring extinction; learning and accountability; biodiversity conservation

**Corresponding author:**
Phillipa C. McCormack;
Email: phillipa.mccormack@adelaide.edu.au

# Implications of extinction in law: Preventing, declaring and learning from species extinctions

## Phillipa C. McCormack

Adelaide Law School, The University of Adelaide, Adelaide, SA, Australia; Law School, University of Tasmania, Hobart, TAS, Australia and Centre for Marine Socioecology, Taroona, TAS, Australia

## Abstract

Biodiversity laws around the world differ, but, at their core, these laws promote the fundamental objective of preventing environmental decline and species extinctions. A variety of legal mechanisms have been implemented in domestic laws around the world to achieve this objective, including protection for habitat, environmental impact assessments and threatened species recovery plans. In many jurisdictions, if these mechanisms fail to protect a species, it may be legally declared extinct, or added to a formal list of those that have been lost. This article examines the conservation purpose and legal implications for laws about extinction. A legal power to recognise a species as extinct has the potential to foster ambition, transparency and rigorous measurement of progress against conservation goals. However, in practice, efforts to prevent extinction are applied selectively. Without an obligation to learn from extinctions, recognition of species extinctions in law may have perverse effects, or no effect at all. This article proposes a conceptual model for the role of law in relation to extinctions, highlighting opportunities to improve legal frameworks to achieve more productive and positive conservation outcomes, even as climate change and other pressures drive many more species towards extinction.

## Impact statement

It is important to understand the role for existing biodiversity laws in relation to extinction before making a case for their reform or improvement, to better prevent extinctions. This article undertakes that task for the first time. It makes the case that these laws reveal two important roles: seeking to prevent extinction and noting the fact of extinction when it occurs. However, this research makes the case that biodiversity laws could readily, and should certainly, also meet a third, crucial role: to facilitate learning when extinctions occur.

## Introduction

Biodiversity laws create a framework of ambition, authority and responsibility to protect nature and help it thrive. At least, that is how it is supposed to work. In reality, shortfalls in ambition, fuzzy legal objectives, and limitations in resourcing, implementation and enforcement have contributed to dramatic biodiversity declines across the world and many species extinctions (e.g., IPBES, 2019; Akhtar-Khavari et al., 2021; IUCN, 2022). Conservation outcomes over recent decades would certainly have been worse without existing legal and policy frameworks (e.g., Greenwald et al., 2019; Bolam, 2021; Rodríguez et al., 2022). However, weaknesses in law and policy must be addressed, in both the terrestrial and marine realms, if we are to avoid what the recent Intergovernmental Science-Policy Platform on Biodiversity and Ecosystem Services (IPBES) report reported as approximately 1-million more species extinctions likely over coming decades (IPBES, 2019; Cloutier de Repentigny, 2020). In fact, IPBES has identified 'environmental law and implementation' as one of the five key interventions or 'levers' for generating transformative change, fundamental to our efforts to tackle the complex underlying drivers of biodiversity loss (IPBES, 2019, D2).

International, regional and domestic biodiversity laws implement broad conservation-oriented goals through legal tools such as: the power to set aside and manage land and water for conservation purposes ('protected areas'); processes for identifying and listing species and ecological communities threatened with extinction and planning for their recovery; processes such as environmental impact assessments, to assess and mitigate the environmental impact of threatening processes such as land use change from mining and agriculture; and obligations to prepare plans for species recovery and to prevent threatening processes such as the spread of invasive alien species. Despite some successes, these laws have not generally been successful at conserving biodiversity (Greenwald et al., 2019; IPBES, 2019). Even the most-developed countries, which are comparatively well-equipped to resource and fully implement biodiversity laws,

have seen high rates of biodiversity loss (e.g., Waldron et al., 2013), and this trajectory of decline is expected to escalate over coming decades to the detriment of human well-being and all life on Earth (IPBES, 2019; Turnhout and Purvis, 2021).

There is a wealth of research about the kinds of legal reforms that will be required to better conserve biodiversity and reverse current trends in biodiversity decline. Recommendations for reform include clearer and more effective protection for species habitat (Venter et al., 2014), stronger protections for ecological communities and ecosystem health and functioning (e.g., Oliver et al., 2004; Beier and Albuquerque, 2015), a clearer focus on climate adaptation in the conservation of species and ecosystems (e.g., McCormack, 2018; McDonald et al., 2019), reduced discretion for decision-makers and stronger enforcement mechanisms, more effective 'mainstreaming' of biodiversity protections across other regulatory frameworks (e.g., Rounsevell et al., 2020) and better resourcing including for long-term biodiversity monitoring across the globe (e.g., Schmeller et al., 2017). Rapid and substantial improvements in resourcing, implementation and enforcement of biodiversity laws are certainly necessary. However, this perspective piece tackles a narrower issue, focusing specifically on the intersection between extinction and law.

Existing biodiversity laws have been implicated in hastening the extinction of some species and failing to avert the loss of others (Woinarski et al., 2017; Cloutier de Repentigny, 2020; cf. Greenwald et al., 2019), but legal research focusing specifically on extinction is limited, at best. Perhaps this reflects a presumption that preventing extinction is the obvious basis for conservation laws; or perhaps it is, as Lim has argued, a reflection of our 'denial of grief as we attempt to distance ourselves from the unimaginable as it occurs before our eyes – and at our doing' (Lim, 2021, 623). In a rare exception to this scholarly gap, Akhtar-Khavari et al. (2021) recently noted that, '[f] or environmental law, extinction is both a pragmatic and technical consideration and a critical and moral compass against which to measure law's effectiveness and its ethical dispositions' (494).

Nevertheless, remarkably little research has sought to articulate or characterise the role that laws about extinction should be playing. This article tackles that narrow question, proposing that laws about extinction have three core roles: to prevent extinction, to acknowledge extinction when it occurs and to provide mechanisms and processes that support learning from species loss. To that end, Part 2 introduces a novel conceptual model that illustrates these three roles for laws about extinction, and explains how the concept of extinction is currently incorporated into biodiversity laws. The article concludes in Part 3 with a call to 'close the circle' by ensuring that biodiversity laws help us to learn from extinction, and help us to more effectively take responsibility for nature and its flourishing, in a rapidly changing world.

## Extinction in biodiversity law

The concept of extinction is not mentioned in most international biodiversity laws – including the core biodiversity conservation agreement: the Convention on Biological Diversity ('CBD') – nor is it mentioned in many regional biodiversity laws, such as the European Union's Habitats Directive.[1] Extinction is more likely to be mentioned in domestic biodiversity legislation; though even at that scale, explicit objectives to prevent extinction do not seem to be common. Even so, the goal at the heart of biodiversity laws at all scales is to arrest biodiversity decline and work towards the recovery of the natural world. This perspective article proposes three critical roles for law in relation to species extinctions. First, biodiversity laws seek to prevent

species extinctions. Second, these laws should (and often do) provide for a declaration or formal acknowledgement of extinctions when they occur. Third, laws should (but typically do not) oblige and empower decision-makers to draw lessons from species extinctions that change and improve conservation management. These three roles are illustrated in Figure 1.

Existing biodiversity laws engage with the concept of extinction in a range of different ways. For example, extinction may be referenced in overarching legal goals, which are not enforceable themselves but demonstrate what is valued and what is sought to be achieved by the implementation of a legal instrument such as a convention, statute or rule. Biodiversity laws also empower, and may mandate that, decision-makers act in particular ways to prevent extinction, including by maintaining lists of threatened species and prohibiting activities that threaten or cause harm to those species (including activities that may cause extinction). Biodiversity laws may also establish institutions or agencies that collect and/or report on important information about the status of species and trends in conservation, providing an evidence base for decisions to prevent extinction and promote species recovery.

Biodiversity laws operate at international, regional and domestic scales, but they are framed and implemented differently at each scale. For example, most international biodiversity laws impose procedural obligations, requiring parties to 'make national plans' or 'ensure processes are in place' to conserve biodiversity. These laws are also consent-based, so governments have to agree (and cannot generally be compelled) to participate in a tribunal or arbitration process if a dispute arises (see Birnie et al., 2021). The content of international biodiversity laws generally becomes enforceable when it is adopted (sometimes verbatim) and implemented through regional and domestic laws. For example, the CBD requires State parties to 'identify and monitor components of biological diversity' that are important for conservation, paying particular attention to those 'requiring urgent conservation measures' (Article 8). One way that this obligation is implemented and enforced around the world is through national laws requiring governments to establish threatened species lists (e.g., 'Domestic Law'; Table 1). Protection of threatened species from extinction is then enforced by governments through, for example, obligations on decision-makers to facilitate the recovery of listed species, and prohibitions on harming or trading listed species, with penalties for non-compliance. In some parts of the world, regional laws also create enforceable obligations to protect species from extinction and other environmental harms (e.g., 'Regional Law'; Table 1).

Table 1 provides a list of indicative, though far from exhaustive, examples of how the concept of extinction is articulated or implied in legal instruments at international, regional and domestic scales in different parts of the world. Table 1 demonstrates that the third important role for biodiversity law – to ensure, through explicit obligations, that we learn from extinctions and avoid replicating past mistakes in future conservation management – is all but absent in legal instruments.

While international biodiversity laws play an important role in articulating ambition and setting conservation standards, the analysis that follows focuses primarily on regional and domestic laws, because that is the scale at which laws about extinction are most commonly implemented, enforceable and amenable to reform.

### The preventative role of law

When operating at their best, laws that seek to prevent extinction ought to foster ambition in conservation management, precaution in

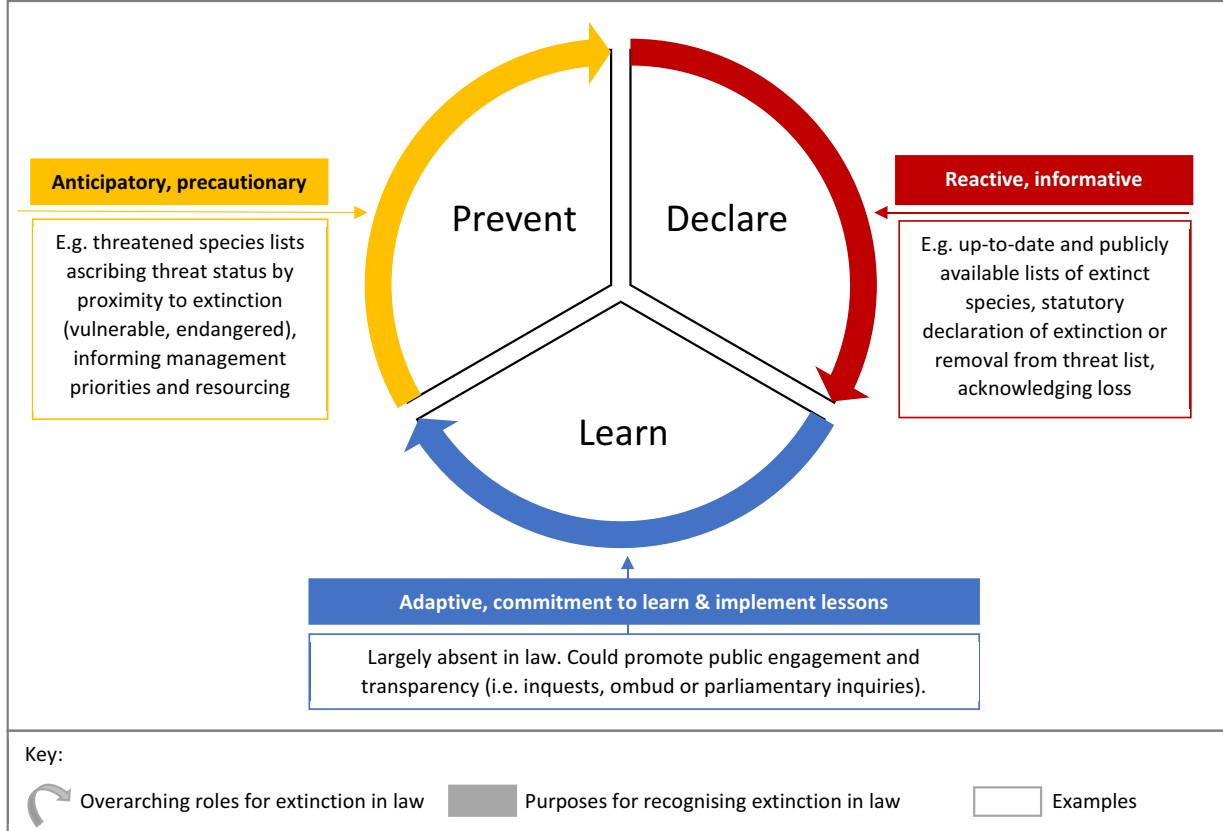

**Figure 1.** The roles and purposes for biodiversity laws about extinction

activities that may harm biodiversity and rigorous measurement of progress against conservation goals. Anticipating the dire possibility of losing an entire species to extinction, preventative mechanisms in law ought to focus our attention on species that are most at risk, while maintaining a focus on the broader goal of conserving abundant species, healthy landscapes and resilient ecological functions.

The first aspect of these laws for preventing extinction is the overarching legal goals that inform the way that biodiversity laws are interpreted and implemented. These goals are sometimes described as purposes, objectives and objects clauses, but their task is to express the primary aims of biodiversity laws against which the success (or failure) of these laws should be measured (McCormack, 2018). Setting an overarching legal goal to prevent extinction is also important because it creates a clear mandate to implement actions that help to achieve that goal. Most biodiversity laws seek to protect nature and avoid biodiversity loss and ecological collapse including (though often implicitly) as a result of extinction. However, some national statutes are very specific about their goal to prevent extinction. For example, the explicit purpose of the *Species at Risk Act 2002* (Canada) is 'to prevent wildlife species from being extirpated or becoming extinct' (section 6). Similarly, as noted in Table 1, the United States' Endangered Species Act begins by declaring that 'economic growth and development' has caused the extinction of 'various species of fish, wildlife and plants' in the absence of 'adequate concern and conservation', and the Act is specifically designed to provide a means to conserve those species threatened with extinction and the ecosystems upon which they depend (section 2).

The second aspect of laws for preventing extinction is the authority and obligations that they create – substantive legal tools

– for identifying and listing species that are threatened with extinction and promoting their protection and recovery. Biodiversity laws create specific rules about the conservation and management of species that are listed on threatened species lists, often requiring governments to prioritise resources for the species that are closest to extinction, identify and protect habitat that is critical to their survival, and prohibit activities that may harm listed species or their habitat without explicit approval. Listing processes required by international laws such as the Convention on Migratory Species (CMS) (see Table 1) are reflected in obligations to list threatened species in regional and national laws, including to protect those species listed in annexes to international conventions.

The effectiveness of threatened species lists for preventing extinction has been the subject of strident critique (e.g., Possingham et al. 2002; Dorey and Walker, 2018; Cardillo et al., 2023). These critiques highlight, among other things, biases that favour the conservation of charismatic species, especially mammals and birds, and the dramatic underrepresentation on these lists of other aspects of biodiversity such as fungi, insects and plants (but see Turnhout and Purvis, 2021; Vietnam's Biodiversity Law; Table 1). Biodiversity that is not listed is typically not prioritised for conservation effort, and, as a result, listing biases have important implications for how we 'count' and seek to avoid extinction (e.g., Scheele et al., 2018; Lim, 2021; Woolaston and Akhtar-Khavari, 2021). Threatened species lists are also criticised for failing to acknowledge and protect the full complexity of nature, focusing resources on 'last ditch' efforts to preserve marginal populations, sometimes at the expense of crucial keystone and abundant species that underpin critical ecological processes (e.g., Pascual et al., 2021; cf. Vietnam's Biodiversity Law; Table 1; Tatarski, 2020).

**Table 1.** Examples of laws about extinction at international, regional and domestic scales

| Role (Fig. 1) | Legal mechanism (goals, obligations, information, institutions) | Comment |
|---|---|---|
| **INTERNATIONAL** | | |
| *Convention on Biological Diversity 1992 (CBD)*[2] | | |
| Prevent | *Overarching legal goal*: conserve biodiversity (implicit: from extinction). *Legal obligations*: conserve, monitor and restore biodiversity; ensure parties adopt plans, policies and measures to achieve the same; monitor/address threatening processes (implicit: to avoid extinction). *Policy goals*: UN SDGs Target 15.5; Aichi Target #12 and Global Biodiversity Framework Target #4 (to 'ensure urgent management actions to halt human induced extinction…[and] significantly reduce extinction risk'). | There is no explicit mention of extinction in the Convention's text. Multiple policy instruments that support the implementation of the CBD do focus explicitly on preventing species extinctions, but these policies are not enforceable and are less likely to be directly implemented in domestic law than convention provisions. |
| Declare | *List of extinct species*: IUCN Red List. *Information institutions and evidence about extinction*: IUCN Species Survival Commission; subsidiary technical advisory bodies (CBD arts 23 and 25). | These bodies cannot compel action but can identify and report on extinctions. |
| *Convention on Migratory Species 1979 (CMS)*[3] | | |
| Prevent | *Overarching legal goal*: conserving migratory wild animals (implicit: from extinction). *Legal obligations*: maintain lists of threatened migratory species to protect species 'in danger of extinction throughout all or a significant portion' of their range (appx I); and migratory species that would benefit from international cooperation, incl for their survival (appx II). *Explicit recovery obligation*: 'range states' for a listed migratory species must, among other things, seek 'to conserve…habitats of the species which are of importance in removing the species from danger of extinction'. | Preventative activities under CMS include a strong emphasis on protecting and restoring important habitat and removing obstacles to migration for migratory species, across their migratory range, especially across international borders, to reduce extinction risks. |
| Declare | *List of extinct species*: no. *Information institutions and reporting*: Scientific Council reports to conferences of the parties on the conservation status of migratory species (CMS art VIII) (implicit: incl if a migratory species becomes extinct). | The Council evaluates the conservation status of migratory species and reports on measures for improvement. |
| **REGIONAL** | | |
| *EU Directive on the Conservation of Wild Birds 2009/147/EC (Birds Directive)* | | |
| Prevent | *Legal obligations*: member states must protect listed bird species and their habitats to ensure that they survive (arts 1–5; annex I); member states are required to encourage 'research and work' as a basis for protecting listed species (art 10), particularly matters including 'national lists of species in danger of extinction or particularly endangered species…' (annex V(a)); member states may introduce stricter protections than the Directive requires (art 14). The obligation to protect species habitat exists *before* a risk of species extinction has materialised (see decision, *CEC v. Ireland* [2002] ECR I-05335). | In evaluating whether a species is in danger of extinction, the Birds Directive requires that decision-makers take into account 'trends and variations in population levels' (art 4). The Birds Directive is enforceable against member states, and its operation is overseen and interpreted by the European Court of Justice. |
| Declare | *List of extinct species*: no. *Information institutions and arrangements*: member states are required to encourage research, and send the European Commission 'any information required to enable it to take appropriate measures for the coordination of the research and work referred to in paragraph 1' (art 10(2)), including in relation to 'national lists of species in danger of extinction' (art 10(1), annex V). | This mechanism creates a regional reporting process for threatened species (member states must maintain their own national threatened species lists) and supports member states to volunteer research about drivers of extinction at national scales. |
| Learn? | Every 6 years, member states must report to the Commission on the status and trends of wild bird species, threats and pressures, and conservation measures taken under the Directive (art 12). | There is no explicit obligation to report on extinction, but this obligation could accommodate such reporting. |
| **DOMESTIC/NATIONAL** | | |
| *Biodiversity Law 2008 (Vietnam)* | | |
| Prevent | *Overarching goal*: 'biodiversity conservation', which includes 'the protection of the abundance of natural ecosystems which are important, specific or representative…[and] the rearing, planting and care of species on the list of endangered precious and rare species prioritized for protection' (art 3) (implied: from extinction). *Legal obligations*: governments must list and protect (implied: from | This Vietnamese law is particularly interesting. The Government has strengthened biodiversity laws in recent years, significantly increasing penalties for harming biodiversity. This law is also unusual as it seeks to protect abundant not just threatened biodiversity; explicitly requires adequate funding for monitoring and data collection; protects buffer zones and ecological corridors for |

*(Continued)*

**Table 1.** (*Continued*)

| Role (Fig. 1) | Legal mechanism (goals, obligations, information, institutions) | Comment |
|---|---|---|
| | extinction), 'endangered precious and rare species prioritized for protection' (art 7); funding for surveys and building biodiversity databases is mandated (art 5(2)); legal protection for animals, plants *and* endemic or valuable fungi and microorganisms from extinction is explicit (arts 47 and 49). *Reporting requirements*: conservation area managers must report every 3 years on the status of biodiversity in their area, including the 'actual status', and plans for conserving, endangered/rare species (art 33). | species movement; and includes a strong focus on ecological restoration. There is no explicit obligation to report on or learn from extinctions, but this regular reporting requirement *could* be amended to support learning. |
| Declare | *List of extinct species*: no. It is not clear how a species that has been listed (i.e., as endangered and precious or rare) will be dealt with if it becomes extinct. | An extinct species could, presumably, be removed from the endangered list if evidence was presented of that fact. |
| *Endangered Species Act 1973 (United States of America)* | | |
| Prevent | *Legal goal*: responding to past declines and extinction of 'various species of fish, wildlife and plants' in the absence of 'adequate concern and conservation' by providing a way to protect them and the ecosystems they rely on (s 2). *Legal obligations*: listing endangered species with 'up-listings' to reflect an increased level of threat of extinction; obligation to plan for recovery and protect habitat critical to survival (s 4); prohibiting activities that increase threat to listed species. *Reviews*: the status of all listed species must be reviewed at least every 5 years to determine if any changes are needed (requiring recovered/extinct species to be identified (s 4(2)). | The U.S. Act is an example of a particularly strong, enforceable domestic law. Critical habitat must be identified at the time of listing, and NGOs and others can litigate a failure to list or up-list a species or to protect habitat. There is no explicit obligation to report on/learn from extinctions, but review requirements *could* support lessons. |
| Declare | *List of extinct species*: no, but a species can be removed from the endangered list, incl if it becomes extinct (s 4). | USFWS (among others) can make rulings about conservation, including to identify a species that is extinct. |
| *Environment Protection and Biodiversity Conservation Act 1999 (Australia)* | | |
| Prevent | *Legal goal*: among other things, the Act seeks to 'protect native species and in particular prevent the extinction, and promote the recovery, of threatened species' (s 3(2)(e)). *Legal obligations*: maintain statutory lists for species that are vulnerable to extinction, endangered, extinct and extinct in the wild (ss 18 and 178); prohibits actions that harm to a listed species or ecological community without a permit (s 19); requires development of conservation advices (s 266B) and allows recovery and threat abatement planning (e.g., s 269AA). *The Australian Govt is currently drafting legislation to replace the EPBC Act. A draft bill is expected by Dec 2023. | The Australian Act is a particularly weak example of biodiversity protection law, with limited enforceability (see discussion below). However, the EPBC Act empowers govt to list ecological communities at risk of extinction; and in 2022, the Australian Govt published the *Threatened Species Action Plan: Toward Zero Extinctions* that sets a goal of preventing any new extinctions of plants or animals over the next decade. |
| Declare | *List of extinct species*: yes. The Minister may list, or re-categorise a listed, species into the extinct, or extinct in the wild, category (s 178(1)). | There is no subsequent obligation, consequence or reporting requirement if a species is listed as extinct. |

Another key critique is the way that laws juxtapose statutory lists for protection with assessment and approval pathways for activities that accelerate extinction trajectories such as land clearing for agriculture and urban expansion. Threatened species lists are also often reactive to existing threats and habitat requirements, which will be insufficient as climate change drives changes to species distributions and habitat availability, both for currently listed and newly threatened species. Legal scholars have highlighted the need for law and governance to move beyond technical legal tools for preventing extinction, such as threatened species lists, while also acknowledging an expanded range of values, worldviews and cosmologies, including the long histories of First Nations peoples with species extinctions and environmental change (Akhtar-Khavari et al., 2021, 495).

Despite these critiques, obligations under conservation laws can be effective if they are strongly enforced by government and/or civil society. For example, the U.S. Endangered Species Act has been unusually effective, avoiding hundreds of extinctions and recovering almost 40 species that have listed and then removed from its threatened species list (Taylor et al., 2005; Evans et al., 2016; Greenwald et al., 2019). The U.S. Act has also been controversial, perhaps because its effectiveness has often been at the expense of competing economic and industry objectives (Taylor et al., 2005; Greenwald et al., 2019). By comparison, Australia has one of the worst extinction rates in the world, and more than 2,000 species are registered on the threatened species list established under the national *Environment Protection and Biodiversity Conservation Act 1999* ('EPBC Act'; Cresswell et al., 2021). Very few Australian species have been recovered to the point where they could be de-listed (Cresswell et al., 2021; Woinarski et al., 2023). A crucial difference between the laws in these two jurisdictions is the near-absence of compellable powers in Australia. That is, decision-makers under the EPBC Act can choose whether to list a species as threatened, whether to protect habitat critical to its survival or develop recovery plans and whether to identify threatening processes and act on them. The Minister must approve a 'conservation advice' for every listed threatened species, explaining why the species has become threatened and either setting out appropriate steps to prevent its further decline *or*, yes, that is *alternatively*, it must set out the steps needed to support the species' recovery (section 266B(2), EPBC Act). However, a conservation advice is a weak legal tool because, provided a decision-maker has 'had regard to' any relevant

conservation advice, the decision-maker can choose to permit activities that are *wholly inconsistently* with that advice (e.g., section 139, EPBC Act; Samuel, 2020).

The third aspect of biodiversity laws for preventing extinction is the institutions and reporting processes established to gather evidence about a species' threat status and trajectory. International scientific information and advisory bodies such as the Subsidiary Body on Scientific, Technical and Technological Advice (under art 25, CBD) and the Species Survival Commission of the International Union for the Conservation of Nature (IUCN SSC, 2021; Maggs et al., 2022; Rodríguez et al., 2022) play an important role here, advising not just international convention secretariats but also producing guidelines and resources that can be used at domestic levels to inform the development and implementation of biodiversity laws, including with tools such as the IUCN's Red List of Threatened Species (IUCN, 2022). Similar bodies also exist at national scales such as, for example, the Canadian Endangered Species Conservation Council (under the *Species at Risk Act 2002*). Biodiversity laws at regional and national scales are also informed by subsidiary strategies and targets adopted by parties to international conventions, such as the Aichi Targets developed under the CBD (e.g., Target 12 sought to 'prevent extinctions of known threatened species' by 2020) and its replacement, the Kunming-Montreal Global Biodiversity Framework. Target 4 of the Global Biodiversity Framework requires parties to the CBD to take urgent action to, by 2030:

> halt human induced extinction of known threatened species and for the recovery and conservation of species, in particular threatened species, to significantly reduce extinction risk.[4]

International targets, including to prevent extinction, are given quasi-legal status through their adoption in formal decisions of the conferences of the parties (e.g., COP15 Decision 15/4), but there are no legal implications for a national government failing to implement or achieve those targets. That is probably one reason why none of the Aichi Targets was achieved in full, despite activities in pursuit of those targets achieving some conservation successes (Bolam, 2021). Including an extinction target in the Global Biodiversity Framework is an important step to help reverse ongoing biodiversity losses (Maron et al., 2021; Williams et al., 2021; Maggs et al., 2022), but the achievement of that target will depend on greatly increased resourcing for conservation activities and a clearer commitment by governments around the world to close accountability loopholes and avoid the trade-offs between conservation and environmentally damaging activities that have been so significant in ongoing biodiversity losses (IPBES, 2019).

### The declaratory role of law

The second role for law, represented as the 'Declare' segment in Figure 1, is a formal process for recognising that a species has become extinct. There is no express declaratory role in any international biodiversity law, which is perhaps unsurprising, given that international instruments are typically implemented at domestic scales through domestic laws. However, the IUCN Red List of Threatened Species, which is not technically a legal mechanism, does include categories for species anywhere in the world that have become 'extinct' or that are 'extinct in the wild' (IUCN, 2022). As such, the IUCN Red List does support an international declaratory function for biodiversity laws.

At the domestic scale, this role for law in declaring or recognising extinctions can take the form of a statutory declaration of extinction and/or re-categorising species from threatened to extinct in a formal list; or it may involve a legal process for removing an extinct species from a threatened species list (see Table 1). For example, the Australian EPBC Act establishes a statutory list with categories for species that are extinct and extinct in the wild, and the relevant Minister may list, or re-categorise a listed species, including into the extinct category (EPBC Act s178(1); see also Species at Risk Act (Canada) s15(1)). Section 4 of the U.S. Endangered Species Act empowers the Secretary to remove a species from a threatened or endangered list, including if it becomes extinct (Table 1).

This declaratory role for law is closely intertwined with the law's role for preventing extinction. The possibility of a species being formally recognised as extinct frames the law's protective provisions as critical for avoiding the outcome of extinction. The power to declare something extinct also creates an obligation to acknowledge that preventative conservation efforts have failed in a particular case. For example, the EPBC Act in Australia provides that a 'native species is eligible to be included in the extinct category… [if] there is *no reasonable doubt that the last member of the species has died*' (section 179(1), emphasis added). Evidence must be provided to support that finding, which demands a close assessment of the species' recorded distribution and potential habitat to determine its absence. Governments and broader communities are called upon to record and acknowledge the fact that a component of the Earth's natural heritage has been lost forever.

There are important, practical limitations on the effectiveness of this declaratory role for law. For example, there can be significant time lags between a species' extinction and a formal, legal recognition of that fact. The ivory-billed woodpecker (*Campephilus principalis*) provides a particularly useful, and controversial, example of this issue. The United States Fish & Wildlife Service (USFWS) records the last 'commonly agreed upon sighting' of the bird as April 1944, but it was not listed as endangered until 1967 (USFWS, 2022). In September 2021, 77 years after that last agreed upon sighting, the USFWS proposed a ruling to de-list the bird, and formally recognise it as extinct. However, following strong public opposition and allegations of a disagreement among experts, the USFWS re-opened public consultation on the proposed ruling and then announced a 6-month extension on its final delisting decision (USFWS, 2022). The agency provided this additional consultation period despite the fact that more than US$20.3 m in public funding and an estimate of over 578,000 h spent over recent decades searching for the bird had failed to yield indisputable proof of its persistence (Troy and Jones, 2023). The USFWS still does not appear to have issued a final ruling on the extinction ivory-billed woodpecker.

This example demonstrates the complexity of this aspect of law which, aside from potentially lengthy delays, may include the possibility that an agency will *never* be able to demonstrate, with incontrovertible evidence, that some species are indeed extinct. Nevertheless, these challenges need not be fatal to the utility of this role for law. Unlike the ivory-billed woodpecker, some species can be demonstrated to be extinct with a high level of certainty. For example, the Brambles Cays Melomys lost all of its habitat on the only island it inhabited as a result of sea level rise and storm surges, and its extinction was tragically clear (Woinarski et al., 2017). Where a species can be identified as extinct, this declaratory role for law can help to raise awareness of biodiversity loss and can prompt conservation concern. Moreover, rapidly improving technology for monitoring and surveying populations, even in remote areas, may allow the status of many more species to be recorded over coming years, and extinctions to

be identified faster and with more certainty. There is value in having a legal requirement to formally recognise that a species has been lost, even if it cannot be used in every potential case of extinction.

Finally, this declaratory role for law should be more than simply a warning tool and a record of loss. It should create a formal opportunity to take notice and even to grieve. Extinction is more than the loss of a single life; it is the end of a genetically and biologically unique component of a system and landscape, and can be the end of a story, songline or relationship. At present, extinction is typically the end of the legal process for a species, but this is an inappropriate conclusion to the role that law might play in helping to prevent extinctions, marking their occurrence, but also learning from their loss. As climate change renders conservation more complex and less certain, biodiversity laws should ensure that an event as serious as a species' extinction becomes a formal trigger for reflection, learning and adaptation.

### The role for law in supporting learning

There are very few formal obligations to draw lessons from conservation outcomes generally, to share those lessons publicly, and to embed learning through changed management practice. Notable exceptions include the obligations for regular, public reporting under the EU Birds Directive (Table 1) and the requirement in some jurisdictions to report on species recovery (e.g., section 4(f)(3) of the U.S. Endangered Species Act requires the Secretary to report every 2 years on the implementation of recovery plans [among other things] to the relevant U.S. House of Representatives and Senate Committees, though lessons identified in that process are not necessarily required to be transferred or adopted by other recovery programmes). This perspective piece focuses on extinction in law, but there is a strong case for emphasising learning across biodiversity laws more generally.

Remarkably, given the significance of a global extinction, few if any legal consequences currently flow when a species becomes extinct (Woinarski et al., 2017; Scheele et al., 2018). An explicit requirement to investigate and actively learn from a species extinction does not appear to be imposed in any existing domestic biodiversity law. Nevertheless, biodiversity laws could, and should, mandate a process of purposeful, transparent and systematic learning after every species extinction (Figure 1). Such a mechanism would reflect the seriousness of extinction while simultaneously helping to ensure that we improve our capacity to avoid extinctions in future (Woinarski et al., 2017; Scheele et al., 2018).

A mandatory learning process would help to avoid complacency in conservation because decision-makers will know that they will be called upon to explain their decisions in the event that a species becomes extinct. Other potential perversities could also be avoided. For example, if an holistic review of species management was guaranteed to follow an extinction, decisions about prioritising resources to (or away from) expensive and challenging conservation efforts for critically endangered species, are more likely to be rigorous, well-justified and based on the best-available science, because there is a real possibility that such decisions will be examined and the results of the review, reported (an argument that is strongly supported by research on administrative integrity more generally, e.g., Brown, 2018). That is not to say that resources should *never* be prioritised away from a critically endangered species, but rather that such decisions should be carefully considered and justified.

Reviewing decision-making and drawing lessons from exceptional circumstances are the bread and butter of legal institutions such as courts, tribunals, commissions and inquiries, as well as integrity bodies such as auditors-general, ombudspeople and, at least in countries with parliamentary systems, parliamentary committees. Empowering a pre-existing institution to conduct a formal inquiry whenever a species becomes extinct, or establishing a new integrity or review body for this particular task would be relatively straightforward. Experience in jurisdictions around the world with various integrity systems mean that we already know the kinds of characteristics that would be necessary for such a body. These characteristics should include independence from government, ideally mandating public or parliamentary reporting rather than reporting to a Minister or Department; secure and adequate resourcing; and sufficient power to call and compel witnesses, subpoena documents and make important findings of fact about, for example, the trajectory of a species' decline and likely contributors to its extinction (e.g., Appleby, 2017; Brown, 2018).

In a detailed analysis of three 'predictable and probably preventable' species extinctions in Australia, Woinarski et al. (2017) demonstrated the power of a coronial inquest style of review for identifying extinction drivers and lessons for future decision-making (13). The authors argue that this style of '[r]etrospective and systematic analysis' could be readily adapted to support a rigorous investigation of species extinctions, in the same way that a coroner is appointed to examine unexpected or unexplained human deaths (Woinarski et al., 2017, 14). A coroner is an independent judicial officer with security of tenure, independence from government, and all of the necessary skills and legal power to investigate and report on the circumstances leading to a human death. Coronial inquests for species extinctions would assess the circumstances of the extinction and, like existing coroners, make findings of fact and recommendations to support improvements in governance standards and practices. Importantly, coronial inquests do not attribute liability to individual decision-makers, and pursuing 'accountability' as a form of blame for extinctions is not what is being proposed here. A liability-oriented approach would almost certainly have perverse outcomes, creating incentives for risk-aversion, bureaucratic blame-avoidance or blame-shifting (including refusal to acknowledge a potential species extinction), and barriers to transparency (see, generally, Hood, 2014).

Legal support for learning from extinction would enhance both preventative and declaratory roles for law. For example, mandatory systematic reviews could generate lessons for improving recovery planning for threatened species, and empower conservation managers to remedy the failings that caused or contributed to a past extinction. Implementing recommendations from an extinction inquest would also help to improve the operation of existing laws for preventing extinction, and may trigger legal reform if existing laws are deemed to be insufficient. The declaratory role for biodiversity laws would also be given greater purpose if it was reimagined as a formal trigger for an independent review of extinction.

### Conclusion: Completing the circle

This article has illustrated the preventative and declaratory roles for biodiversity law in the context of species extinction. These two roles are important but not sufficient to support effective conservation. Existing shortfalls and perversities in biodiversity laws must be improved if we are to transform our relationship with nature (IPBES, 2019), but, to do so, biodiversity laws need to 'complete

the circle'. To achieve that outcome, biodiversity laws must be equipped with a coherent accountability mechanism that fosters learning from species extinctions, so that decision-makers can identify and respond to systemic and ongoing drivers of extinction even as the climate changes.

Negotiations are currently underway on the implementation of the Global Biodiversity Framework, designed to guide implementation of the CBD over the coming decade. These negotiations have implications but also opportunities to improve conservation outcomes at regional, national and sub-national scales (e.g., Perino et al., 2022). This perspective piece argues that commitments to prevent extinctions, such as Target 4 of the new Global Biodiversity Framework, are important, but must also be supported by legal powers to declare or recognise loss when it cannot be prevented. Moreover, biodiversity laws must foster learning, by introducing mandated review processes that ensure decision-makers actively and consistently learn from species extinctions. Completing the circle may yet help us to improve conservation outcomes and take collective responsibility for nature and its flourishing, in a way that is desperately overdue.

**Open peer review.** To view the open peer review materials for this article, please visit http://doi.org/10.1017/ext.2023.19.

**Acknowledgements.** I am grateful to two anonymous reviewers for their constructive feedback, which improved this article a great deal. I am also grateful to Maya Clarke for her valuable research assistance. I take full responsibility for any errors or omissions.

**Author contribution.** I conceived, designed, analysed and drafted the piece in full.

**Financial support.** This research received no specific grant from any funding agency, commercial or not-for-profit sectors.

**Competing interest.** The author declares no competing interest.

## Notes

1. African Convention on the Conservation of Nature and Natural Resources, opened for signature 15 September 1968, 1001 UNTS 3 (entered into force 16 June 1969), as revised in 2003.
2. Convention on Biological Diversity, opened for signature 5 June 1992, 1760 UNTS 79 (entered into force on 29 December 1993) ('CBD').
3. Convention on the Conservation of Migratory Species of Wild Animals, opened for signature 6 November 1979, 1651 UNTS 333 (entered into force 1 November 1983) ('CBD').
4. Conference of the Parties to the CBD, Fifteenth Meeting, 'Kunming-Montreal Global Biodiversity Framework' (Montreal, Canada, 7-19 December 2022) (19 December2022) CBD/COP/DEC/15/4 ('COP15 Decision 15/4').

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
