## [Reviewer Report]

Adding a requirement that the causes of a species extinction be investigated and disclosed to existing laws for protecting biodiversity is novel and interesting. It could help raise awareness about extinction and it’s causes, hopefully leading to increased action to avoid extinction. One problem with this idea that needs to be addressed in the manuscript is the significant delay, frequently decades, between a species' extinction and its official declaration. This delay has important implications related to the utility of the proposed investigations for future species conservation. For example, the ivory-billed woodpecker almost certainly went extinct in the 1940s, but was only declared extinct in 2021 and not without controversy. During the intervening nearly 80 years, the long-leaf pine forests needed by the woodpecker continued to decline and many other species in this ecosystem have since been found to be at risk, including the red-cockaded woodpecker. I would like to see the implications of the time-lag between actual extinction and declarations of extinction discussed in a revised version of the manuscript.

---

## [Reviewer Report]

I think this is a worthwhile paper that aims to clarify the roles of biodiversity laws in protecting biodiversity. It offers much food for thought for conservation biologists whose work is frequently contextualized and guided by such laws. This paper offers a conceptual framework based on three overarching roles for biodiversity law - prevention, declaration, and learning - and highlights mechanisms by which these roles currently are or could be supported in biodiversity laws. I have a few comments, questions, and suggestions for revisions that I hope will clarify aspects of this manuscript.

International, regional, and domestic laws are lumped together for much of the discussion in this paper. However, the effectiveness of laws to protect biodiversity - and their specificity - depends largely on the ability to enforce laws, which relies on other legal mechanisms and incentives which are more likely to be enforceable at smaller scales, i.e. domestic and regional. However, the transboundary nature of many species distributions and the global scale of threats necessitates international treaties to protect biodiversity. But biodiversity laws also have different goals at different jurisdictional scales. Furthermore, most or all international biodiversity treaties do not meet their targets. I think this paper should attempt to define in general terms international biodiversity laws versus domestic or regional laws at the outset and discuss the intentions of and differences between laws across these jurisdictional scales. To that end I think this paper would benefit enormously from a table that includes examples of international, regional, and domestic biodiversity laws highlighting their attributes, e.g. whether and how they encapsulate prevention, declaration, and learning and whether there are mechanisms for enforcement. It does not have to be exhaustive but it would be useful to have a few more structured examples than the small number included in this manuscript and to see how they compare and contrast in the context of the conceptual framework presented here.

In section 2 on page 3, the paper presents “four distinct but related mechanisms” but then immediately moves on to focusing on only three mechanisms presented in Figure 1 (mislabeled as Figure 2) with insufficient explanation or discussion of how the list of four mechanisms relates to biodiversity laws, how they translate to the Figure (as they are called different things), and why they need to be highlighted in the brief way that they are. There is no citation offered for these four mechanisms. This is very confusing. I recommend that this section focus on describing the conceptual framework as it is structured in Figure 1 and provide more context for each of the three components rather than introducing the four mechanisms and then not pursuing them or connecting them to the Figure in a meaningful way.

In section 2.1 it states that “no international biodiversity law makes an explicit commitment to avoid preventable species extinctions” and then goes on to state that the Convention on Biological Diversity (CBD) does not commit parties to the goal of preventing extinctions. First, without a survey of international law - or a citation to one - it is unknown to the reader whether international biodiversity laws are devoid of extinction prevention commitments or targets. Second, the Aichi Target 12 (now superseded by new proposed targets) explicitly commits to preventing extinctions - “By 2020 the extinction of known threatened species has been prevented” - which is acknowledged further on in the manuscript. In the new proposed goals for the Post-2020 Global Biodiversity Framework, Goal A of the “2050 Goals and 2030 Milestones” explicitly states that “the rate of extinctions has been reduced at least tenfold, and the risk of species extinctions across all taxonomic and functional groups, is halved.” And the “Components of the Goals and Targets” of Goal A of the Post-2020 Framework also explicitly state “A.3. Prevent extinction and improve the conservation status of species.” Page 7 of this manuscript goes on to say that the “first draft of the Post-2020 Framework does not include a specific commitment to avoiding extinction” - that is not my reading of the Post-2020 Framework at all, as evidenced by these excerpts. So I think this section of the paper should be revised to include a more nuanced and detailed treatment; I think it glosses over too much by relying merely on the Preamble to the CBD. Also, I didn’t see this paper cited but I think it might be relevant to what the author is trying to get at here: Rounsevell et al. A biodiversity target based on species extinctions, Science 2020.

Section 2.2 deals with the “declaratory role of law” in the manuscript’s conceptual framework. The author may be correct that international laws do not have a declaratory role, but why does it need it? It certainly is the case that when species are declared extinct by government agencies or the IUCN or other conservation NGOs, people know about it through press releases and the mainstream media. I don’t know what a declaratory role would look like for an international law that has no authority of enforcement - moreover, a law can’t make a declaration of extinction but agencies and organizations can and do. For instance, in the United States such declarations are made by the US Fish and Wildlife Service (e.g. see https://www.fws.gov/press-release/2021-09/us-fish-and-wildlife-service-proposes-delisting-23-species-endangered-species). I think this speaks to the different roles of international, regional, and domestic biodiversity laws, and the types of laws which, as pointed out above, needs a deeper treatment in this manuscript. I would also add that declarations of recovery are equally, and possibly more, important than declarations of extinction. It is highly useful to know what actions led to the recovery of species.

Section 2.3 addresses the “role for law in supporting learning” and focuses on the importance of learning why an extinction occurred. I would argue that by then it is too late, the classification of a species as at risk of extinction typically highlights the causes, and that there is a huge role for learning while the species is still extant. This role for learning currently exists in the United States Endangered Species Act by the requirement of recovery plans of the federal agencies responsible for implementing the ESA. And I dare say other laws in other countries also have established mechanisms for learning. I think a focus on what these laws are missing and how they can be bolstered (e.g. by mandating monitoring and providing for a budget to do so) would be more illuminating here. I recommend that this section pay greater attention to the mechanisms for learning that currently exist in specific biodiversity laws and also highlight the role of learning from recovery. Again, a table listing the attributes of different biodiversity laws would go a long way to clarifying the role for law in supporting learning and where gaps currently lie.

Given the demonstrable failure of international laws to prevent extinctions and biodiversity loss (e.g. none of the Aichi targets were achieved by 2020), I am skeptical that the framework in Figure 1 can be successful. While there are no meaningful repercussions for failing to meet targets, they will not be met. Domestic laws will tend to have more “teeth” because they operate within an enforceable legal framework with repercussions and incentives. Even so, these laws - international and domestic alike - are failing biodiversity in large part due to lack of political will in the “Prevention” component of the conceptual framework in Figure 1. I think this paper would be improved by also examining this aspect of where laws fail, what can be (and has been) done to successfully protect biodiversity, and how this intersects with or results from biodiversity laws, with some salient examples.

---

## [Editor Report]

I agree with the two reviewers that this study is well-written, interesting and original in its approach. This review can add valuable ideas to the field of conservation law. The reviewers suggest some areas of clarification, extra discussion and examples. I agree with reviewer one that it would improve the manuscript to briefly discuss the implications of the time difference between likely extinction and the date of official confirmation in publications or on the Red List (or regional lists). Reviewer two has suggested several important clarifications including addressing the different spatial scales of laws and law enforcement, framing the discussion of mechanisms in terms of the scheme in figure 1, clarifications of some of the detail in section 2.1 and the potential roles of law in declarations of species status. Reviewer two suggests focusing more on recovery of threatened species in a section of the paper, however the key focus of the review should be on extinction- I think that any discussion of the importance of learning from extinction vs learning from the trajectories of highly threatened species should not stray too far from the relevance to extinction (including because of the word limit). However, added discussion of why laws fail would work well with this topic.

---

## [Reviewer Report]

I cannot seriously question the idea that when species are found to be extinct, there should be an investigation to determine what went wrong. The unfortunate truth, however, is that in most cases, there is no mystery. The causes of species extinction--habitat destruction, invasive species, etc.--are well known. The only real question is whether we have the political will and wherewithal to address these factors. And thus, should countries or other entities require declarations of extinctions and investigations to determine the cause, the answer is likely to always be the same--we didn’t do enough.

---

## [Reviewer Report]

I have read through this revised version of the manuscript and I believe it has addressed my original comments on a previous version. I have one small requested edit - the caption for Figure 1 should be descriptive not a question.